# It Is Not Just Stress: A Bayesian Approach to the Shape of the Negative Psychological Features Associated with Sport Injuries

**DOI:** 10.3390/healthcare10020236

**Published:** 2022-01-26

**Authors:** Aurelio Olmedilla Zafra, Bruno Martins, F. Javier Ponseti-Verdaguer, Roberto Ruiz-Barquín, Alejandro García-Mas

**Affiliations:** 1Department of Personality, Evaluation, and Psychological Treatment, Campus Regional Excellence Mare Nostrum, University of Murcia, 30100 Murcia, Spain; olmedilla@um.es; 2GICAFE (Research Group of Sports Sciences—UIB), University of Lisbon, 1649-004 Lisbon, Portugal; bruno.pt@gmail.com; 3GICAFE (Research Group of Sports Sciences), Department of Pedagogy, University of the Balearic Islands, 07122 Palma, Spain; 4Department of Evolutive and Educational Psychology, Autonomous University of Madrid, 28049 Madrid, Spain; roberto.ruiz@uam.es; 5GICAFE (Research Group of Sports Sciences), Department of Psychology, University of the Balearic Islands, 07122 Palma, Spain; alex.garcia@uib.es

**Keywords:** sports injuries, psychological factors, football, Bayesian network

## Abstract

The main objective of this study is to extend the stress and injury model of Andersen and Williams to other “negative” psychological variables, such as anxiety and depression, encompassed in the conceptual model of Olmedilla and García-Mas. The relationship is studied of this psychological macro-variable with two other variables related to sports injuries: the search for social support and the search for connections between risk and the environment of athletes. A combination of classic methods and probabilistic approaches through Bayesian networks is used. The study samples comprised 455 traditional and indoor football players (323 male and 132 female) of an average age of 21.66 years (±4.46). An ad hoc questionnaire was used for the corresponding sociodemographic data and data relating to injuries. The variables measured were the emotional states of: stress, depression and anxiety, the attitude towards risk-taking in different areas, and the evaluation of the perception of social support. The results indicate that the probabilistic analysis conducted gives a boost to the classic model focused on stress, as well as the conceptual planning derived from the Global Model of Sports Injuries (GMSI), supporting the possibility of extending the stress model to other variables, such as anxiety and depression (“negative” triad).

## 1. Introduction

Sports injuries have been the focus of many studies in different scientific disciplines, showing that they relate to a complex and multicausal phenomena [1,2,3,4,5]. In addition, their incidence has an important impact both on sports competitions and on the primary health care system, as well as on the athletes themselves. Epidemiological data show high levels of probability of suffering sports injuries, and on many occasions the clinics of sports federations cannot attend all injuries, referring them to the general health system, especially in competitions of young people in training stages [6,7,8,9]. On the other hand, the impact on the injured athlete often compromises his or her emotional and psychological state, in addition to other problems of a sporting, social, and even economic nature [10,11].

In fact, there are several factors related with the injuries’ occurrence, such as biomedical, dispositional, nutritional, or postural ones [12], related to the field or pitch conditions, or with some psychosocial factors other than the practitioners themselves [13]. Another one of the aspects demonstrated in research conducted over recent decades is the role of psychological factors in the risk of athlete injuries [14,15,16,17,18]. The stress and injury model of Andersen and Williams [19,20] shows that stress is a key factor in the origin of injuries. Amid potentially stressful situations in the sporting field, athletes may have a stress response that increases muscle tension, which, in turn, hinders motor coordination and reduces flexibility. Furthermore, the stress response already mentioned also reduces the visual range of athletes, causing a significant loss of peripheral information and increasing distraction levels. The model also assumes that the response to stress is moderated by three main factors: the personality of the athlete, history of stressors, and coping resources. These psychosocial variables may alleviate or aggravate the stress response and may eventually affect the vulnerability of athletes to injury [3,21,22,23,24,25]. Although stress is the cornerstone of the model, other psychological and psychosocial variables also play a significant role (different aspects of personality, history of stressors—including the history of injuries—coping resources), as indicated in the Stress and Injury Model of Andersen and Williams [19]. Specifically, one of the aspects most emphasised is anxiety as a personal trait, which may determine the stress response of athletes [20], including recent expansions of the model, such as the ones linking the stress features with the Self Determination theory (frustration) [26].

Different studies have shown that high anxiety trait levels are linked to the vulnerability of athletes to injury [19,27,28,29]. In the review of Cagle et al. [30], it was found that four out of the six papers analysed identified anxiety as an injury predictor, while the other two did not. In any case, they ratified prior research conclusions [31] indicating that other psychosocial characteristics, including coping resources, worry, irritability, and stress, often seem to combine with anxiety in the prediction of sports injuries.

Furthermore, in addition to the history of stressors of athletes, the coping resources available may reduce the stress-injury link. One of these resources is the social support perceived. The scientific literature points to non-conclusive results when this variable has been studied [17,32,33,34,35]. The inconsistency of these results is probably due to the fact that the mediator role of social support is more complicated and unstable than initially thought [34].

In any case, it seems that a positive psychological disposition, which facilitates good management of competitive stress, may act to curb the risk of injury of athletes. Conversely, a negative psychological disposition may favour and increase vulnerability to injuries. Therefore, the Global Psychological Model of Sports Injuries (GPMSIs, or MGPsLD in Spanish) of Olmedilla and García-Mas [36] proposes theoretically delving deeper into the study of the specific psychological variables of greatest significance in the literature, those considered as such in the model of Andersen and Williams [19,20] and those present in the Conceptual Axis of the GPMSIs: psychosocial stress, coping resources (social support), and risk-related behaviour. Furthermore, other negative psychological variables, such as depression, have been studied in research on injury rehabilitation resulting from serious injuries [37,38,39]. However, there are few studies that consider depression as one of the causal factors of injuries [40,41].

Bayesian networks (BNs) are beginning to be widely used in social sciences [42,43], and were recently presented as a useful methodology in sports psychology, given their ability to provide information on the probability of occurrence of events related to performance in sports or, for example, the likelihood of sports injuries. BNs, also referred to as causal networks or belief networks, are a form of statistical modelling that allows us to obtain a graphical network describing the dependencies and conditional independencies from empirical data. The graphical representation of BNs captures the compositional structure of the relations and the general aspects of all probability distributions that are factorised according to that structure. They have proven to be a promising tool for discovering relationships between negative features in sport [44], and in many other sport-related studies, such as cooperative teamwork, motivation and types of sporting cooperation among players in competing teams, motivational climate and competitive anxiety, psychological variables related to athlete injuries [45], the relative effect of age [46,47,48,49], and the relation between sport and educational performance [50]. In line with our study, a number of papers have recently been published that use a new approach, called dynamic BNs, which strives to predict and then mitigate the probability of injuries occurring in athletes [51]. Therefore, taking into consideration the importance of the incidence of sports injuries in the population of athletes and in the general health system, as well as their psychological impact on the athletes involve, the main objective of this study is to extend the stress and injury model of Andersen and Williams [19,20]—repeatedly verified—to other “negative” psychological variables, such as anxiety associated to competition and depression, encompassed in the conceptual model of Olmedilla and García-Mas [36]. Additionally, as a parallel objective, the relationship is sought to be studied of this psychological macro-variable with two other variables related to sports injuries: the search for social support, and the search for connections of risk with the environment surrounding the athletes. Finally, these variables and their relationships are studied with two specific characteristics: (1) on a wide sample of traditional and indoor football players, with a 50% combination (approximately) of injured and non-injured players; and (2) methodologically, a combination of classic methods and probabilistic approaches through Bayesian networks is used.

## 2. Materials and Methods

### 2.1. Design

The research conducted corresponds to an ex post facto study [52]. Furthermore, it is based on a descriptive and longitudinal correlational design. The independent and predictive variables used are: personal data, socio-demographic and injury records; stress levels; anxiety and depression levels; and social support perceived. As dependent variables for prediction purposes (variable criterion), a record of injuries sustained over the season is used.

### 2.2. Participants

The study sample is a non-random and non-probabilistic sample of an incidental and intentional nature. The number of participants assessed entails that the results to be obtained in the study do not exceed a sample error of 5% (*n* > 400; 95% reliability level) [52].

Regarding sporting data, in Table 1, the mean (X) number of years the participants have been federated for is XYF = 11.83 years (SD = 5.41), with 55.2% having spent at least 12 years as a federated athlete. In terms of training days per week, the mean is 3.66 days/week, SD = 1.18), with a broad range of training days, ranging from 2 to 7.

### 2.3. Instruments

To collect the personal information, an ad-hoc questionnaire on personal and socio demographic data was used. The questionnaire comprised nine questions referring to age, sex, sport played, current club, position played, competitive level, days of training per week, and the duration of daily training sessions.

To collect information on injuries sustained, an ad-hoc self-report on injuries was used. It was a self-reporting questionnaire comprising nine blocks of information to determine the injuries sustained during the previous season. The following questions were included: the number of injuries sustained during the previous season; when the injury happened (month); the time spent out before returning to the sport without discomfort; the specificity of the injury (type, place, detailed description of the injury); establishment of the main causal factors; degree of severity; degree of injury impact on subsequent performance; and casual attribution of the athlete to the injury.

The DASS-21 Questionnaire was used to assess depression, anxiety, and stress. The adapted Spanish version of the DASS-21 Questionnaire of Lovibond and Lovibond [53], undertaken by Román et al. [54], was used. The main objective of this scale is to assess the emotional states of stress, depression, and anxiety. The participants respond through a four-step Likert-like scale (0 = not applicable to me; 1 = slightly applicable to me or a small part of the time; 2 = largely applicable to me or a large part of the time; 3 = very applicable to me or most of the time). Each scale comprised seven items with a score ranging from 0 points (minimum) to 21 points (maximum). The reliability levels obtained by Lovibond and Lovibond [53], with a sample of 717 psychology students, were high or very high: Stress: *α* = 0.89; Depression: *α* = 0.91; Anxiety: *α* = 0.81.

To assess the risk tendency, the DOSPERT-S Questionnaire was used. The adapted Spanish version of the scale created by Weber, Blais and Betz [55], translated by Rubio and Narváez [56] and validated by Lozano et al. [57], was used to assess attitude to risk-taking in different areas. The original and principal scale comprised 40 items. A total of eight additional items were added to the revised scale. However, after an exploratory factorial analysis, it was reduced to 30 items. Furthermore, the range of possible answer options was extended, ranging from five to seven (from 1: extremely unlikely; to 7: extremely likely). The questionnaire comprised five factors, each with six items. The following indicators were observed in this study: Social: *α* = 0.70; Recreational: *α* = 0.80; Finance: *α* = 0.77; Health/Safety: *α* = 0.63; and Ethics: *α* = 0.58.

To assess social support, the Multidimensional Scale of Perceived Social Support of Zimet et al. [58], was used. The adapted Spanish version produced by Landeta and Calvete [59] was used in this study. The original scale [58] comprises 12 items that assess the perception of social support. The response system entails a four-step Likert scale (“almost never = 1”; “sometimes = 2”; “frequently = 3” and “almost always or always = 4”). These 12 items are distributed into three factors. The reliability levels found by Zimmet et al. [58] were: Family: *α* = 0.87; friends: *α* = 0.85 and significant others: *α* = 0.91; in the Spanish version, a seven-step scale was used, which ranged from 1 = completely disagree up to 7 = completely agree, with 4 = neither agree nor disagree).

### 2.4. Procedure

The Regional Football Federation of Murcia (FFRM, Spain) was informed of the study, and both permission from and collaboration with it was requested. Traditional and indoor football teams that met the convenience sampling requirements were recruited. Subsequently, a meeting with the coach, players and parents of minors was scheduled. Finally, the players willing to participate in the research signed an informed consent form. In the case of minors, the form was signed by their parents or guardians. The tests were undertaken on an individual basis. Furthermore, this study was conducted in accordance with the recommendations of the Declaration of Helsinki, and it was approved by the Ethics Committee of the University of Murcia (ID: UM 1551/2017).

### 2.5. Data Analysis

All the standard statistical analyses were conducted using the IBM Corp. (released 2013) IBM SPSS Statistics for Windows, Version 22.0 (IBM Corp., Armonk, NY, USA). Subsequently, a Bayesian network analysis was conducted, making it necessary to determine the structure via a directed acyclic graph (DAG) and to assign conditional probabilities to each node of the DAG. Therefore, learning a BN involves the following two tasks: (i) structural learning, in other words, identifying the topology of the BN, and (ii) parametric learning or estimating the numerical parameters (conditional probabilities) given the topology of the network.

Structural learning was used to obtain the BN through the “BN learn package” [60] using R language [61]. To obtain the structure, the options were to use either a search and score algorithm [62], which assigns a score to each BN structure and selects the model structure with the highest score, or a constraint-based search algorithm [63], which establishes a conditional independence analysis on the data to generate an undirected graph and convert it into a BN using an additional independence test. The score-based algorithm Tabu [62] was used, which was a plausible model for our data, looking for the structure that best improves the score, e.g., using the highest one.

The final model was obtained by repeating structure learning several times (applying bootstrap resampling to our dataset); many network structures were explored (500 BNs) to reduce the impact of locally optimal (but globally suboptimal) networks on learning. The networks learned were averaged to obtain a more robust model. The averaged network structure was obtained using the arcs present in at least 85% of the networks, which gives a measure of the strength of each arc and establishes its significance when given a particular threshold (85%) [64].

The BN is used to make inferences, that is, to calculate new probabilities when new information is introduced [65]. Therefore, after building the BN, some instantiations were conducted (injection of hypothetical variables) to the bottom variables, as well as observation on how the node, bottom, and top variables change their probability values [66].

## 3. Results

With regard to the number of injuries, all the participants assessed ranged from having no injuries (0) to a maximum of five during the previous season, with a mean of 0.59 injuries/season (SD = 0.72), as can be seen in Table 2.

As regards the analyses of injury frequency, Table 2 shows how 51.6% of the sample (*n* = 235) stated they did not have an injury; 39.8% (*n* = 181) stated they had one injury; 7% (*n* = 32) stated they had two injuries; and 1.1% (*n* = 5) stated they had three injuries during the season. It was found that only one athlete had had four injuries, and another had had five injuries during the season (2%, respectively).

Regarding the athletes that stated they had sustained at least one injury (*n* = 220; 48.4% of the total), in the record of the first injury, referring to the month in which the injury took place, 83.6% (*n* = 184) responded. The frequency of injuries per month is as follows (from low to high incidence): October (*n* = 27; 14.7%); February (*n* = 26; 14.1%); November (*n* = 23; 12.5%); March (*n* = 20; 10.9%); January (*n* = 19; 10.3%); September (*n* = 17; 9.2%); May (*n* = 14; 7.6%); April (*n* = 12; 6.5%); August (*n*= 5; 2.7%); and July (*n* = 1; 0.5%). The frequency distribution corresponds almost exactly with the distribution of training sessions and games during the year and the active season.

With regard to time spent out injured before returning to the sport, 151 athletes responded, and the mean number of days without playing was 54.13 (SD = 65.29). The number of days spent out ranged from a minimum of two to a maximum of 420. The frequency analyses show that the number of inactive days in the greatest number of cases is 14 (*n* = 32; 7%), followed by 60 days (*n* = 23; 5.1%), and 30 days (*n* = 22; 4.8%), in that order. Almost half of the sample corresponded to a period under 29 days (45.7%), while the remaining 54.3% spent over 29 days out without playing.

As can be seen in Table 3, comparison between the number of injuries according to the sports category variable per sex, the prevalence rate is greater in the female subgroup (X = 0.62; SD = 0.683) than in the male subgroup (X = 0.58; SD = 0.737), which is slightly lower despite having a greater value dispersion. Analysis of mean differences using Student’s *t*-test shows the absence of statistically significant differences between groups (Levene’s test: F = 0.072; *p* = 0.789; t = −0.567; *p* = 0.571).

Regarding the type of sport, a higher prevalence of injuries was observed in the traditional football group (X = 0.61; SD = 0.718) than in the indoor football group (X = 0.48; SD = 0.738). Analysis of mean differences using Student’s *t*-test shows the absence of statistically significant differences between groups (Levene’s test: F =0.218; *p* = 0.641; t = 1.210; *p* = 0.227).

Considering the age variable, the athletes aged 18 years old or over show a higher prevalence (X = 0.62; SD = 0.730) than the group under 18 years old (X = 0.45, SD = 0.664). Despite not obtaining statistically significant differences, there is a trend towards statistical significance when comparing the two groups (Levene’s test, F = 0.819; *p* = 0.366; t = −1.821; *p* = 0.069; *p* < 0.10). Despite not obtaining statistically significant results in this comparison, Cohen’s d [67] was applied to calculate the effect size for this group. The results show a value of d = 0.244, which is a relatively small value.

Therefore, based on the studies carried out and the results shown in Table 2 and Table 3, we can see that no statistically significant differences have been found (and that the trends have a small effect size) among the various variables in the study. As such, this justifies moving on to the second phase of the study, based on the search for probabilistic relationships through the analysis and generation of Bayesian Networks.

As for the study of correlations between the psychological variables and the epidemiological variables (those of the participants and those of the injury itself), the results obtained are in line with those found in the analysis of statistical differences: No significant correlations were found between the two types of variables.

### 3.1. Graph and Generated Bayesian Network

As can be seen in Figure 1, the resulting graph shows that the antecedent and trigger variable associated with the likelihood of having a sports injury is stress.

Two nodes are clearly formed, one around the psychological variables (stress, anxiety and depression—all three with low probabilities of occurrence in the population studied–the latter being twice as likely as anxiety), and the other with all the risk tendency factors, although they have different probabilities of occurrence associated with the sports injury. One of them, the FJ, acts autonomously (although from low probability values in our population), as an antecedent of this node, i.e., it would seem that the risk tendency in the financial section related to betting, even in the field of sports, would be a trigger for the other tendencies towards risk behaviors (all with low values of occurrence, except two: the risk factors associated with social habit, above all, and secondly, the sporting and/or recreational one.

As for the bottom (or descendant) variables, it is very clear in the BN conducted that Social Support, with a very high probability of occurrence (almost 100%), would not be a predictor factor, but rather would be an almost inevitable consequence of the athletes’ perception given the injury.

Finally, it should be noted that the probability values found in the population studied are generally very low, which is in line with the low prevalence of injuries (around 50% of the sample did not suffer any injury, as can be seen in Table 2). Furthermore, the probabilistic weight of psychological factors is not very high, perhaps having a contributory role rather than a determining one—in terms of the probability of triggering the injury—in the anxiety-stress of the athletes.

### 3.2. BN Validation

The BN was validated using a 10-fold cross validation, taking the area under the curve (AUC), accuracy, sensitivity, and sensibility into consideration. Certain terms should first be defined to understand the validation used: true positive (TP), true negative (TN), false positive (FP), and false negative (FN). If an observation is labelled correctly within its class, it is considered a true positive. Conversely, if an observation is labeled correctly as not belonging to a specific class, it is a true negative. Both TP and TN suggest a consistent result in the classifier.

However, no classifier is perfect and if the model incorrectly labels an observation as belonging to a certain class, it is a false positive; and when incorrectly labelled as not belonging to a certain class, it is designated as a false negative [68]. Both FP and FN indicate that the results from the classifier are contrary to the actual label [59].

Sensitivity, specificity and accuracy are described in terms of these concepts: Sensitivity = TP / (TP + FN); Specificity = TN / (TN + FP); and Accuracy = (TN + TP) / (TN + TP + FN + FP).

The AUC shows that the probability of a randomly chosen positive datum being correctly ranked is much higher than for a randomly chosen negative datum [69]. The readings provide a complete overview of the performance of the BN. As Table 4 shows, the validation tables provided some average results, along with some medium values. These validation values should be considered when undertaking the next step in the Bayesian analysis process (the instantiations) and the final conclusions.

### 3.3. Instantiations

Based on the above results, it was decided to explore, as in previous studies [48,66,70], the changes in the probabilities of occurrence of antecedent variables when hypothetical values are “injected” into bottom or consequent variables. To be more precise, and also considering the meaning of the variables, the instantiations that have been carried out are as follows:(a)Social support, passing to 0% HIGH;(b)DOSPERT-S, Social/Safety, passing to 100% HIGH;(c)Depression, passing to 100% HIGH;(d)The union of the three above, simultaneously.

As can be seen in Table 5, removing the probability of occurrence of Social Support entails: increase in the likelihood of the depression, anxiety and—to a lesser degree– stress triad; decrease in the tendency of risk behavior, albeit small, and, above all, in ethical, social (“go it alone”) and recreational/sporting behavior. In any case, the results of this first instantiation reaffirm those found in the original BN generated.

## 4. Discussion

The results obtained in this study largely confirm the aims expressed in the initial objectives, both from the perspective of the quality of the explanations they allow for and the validity values found. No differences were found between the variables relating to players or the nature of the injuries. Along the same lines, no correlations were found between said variables, which occurred repeatedly when attempts were made to “connect” the objective characteristics of the injury or sport with the psychological variables considered in different studies [36,71].

The information collected in the ad hoc injury questionnaire applied in this research provides valuable information regarding injuries sustained during the season. However, once analyzed—beyond the limitations in its design that will subsequently be discussed–we observe that, once again, no clear pattern emerges that could justify a behavioral approach to the occurrence of sport injuries.

In terms of the probabilistic analysis carried out, consideration should be given, firstly, to providing support to the classical model centered on stress, as well as to the conceptual ordering derived from the Global Sports Injury Model [36], as they can be considered—among several other factors or variables, e.g., related with the characteristics of the practice, the pitch, or the participants’ biomedical features as antecedents to the others. Furthermore, these results—obtained with a methodology based on probability analysis—support the possibility of extending the stress model to other variables (anxiety and depression, the so-called negative “triad”), while maintaining the same characteristics with respect to sport injuries.

Thus, the trigger variable and from which all the other “negative” psychological variables (gathered in a very clear node in the graph obtained) descend is the stress perceived by the athletes. The probability values found are low (it should be remembered that the incidence of injury in the sample studied is only about 50%), and here too stress is pre-eminent in terms of the probability of occurrence. The relatively low probability of depression (although it is the one with the highest value within the BN) and, secondly, of anxiety associated with competition, makes us reflect on whether it is possible to contemplate a component of “relief” from the injury—cease competing—in comparison with the low probability of anxiety found, perhaps related to an early return to competition and its demands. This result had already appeared—in a much smaller form—in a very early study by Liberal et al. [71]. At what point in the natural life of the sports injury does each of these variables carry the most weight? As the study is retrospective, further studies are needed to better understand whether this variable affects prevention (as it seems to do) rather than recovery. Were it to regard learned avoidance behavior, we would be entering more complex territory, which would require observation and a different theoretical approach to those usually carried out [72,73].

It is very interesting to have found that the probability values related to the various factors associated with risk-seeking tendencies (in different domains) constitute another node isolated from the previous node of the “negative triad”. However, this result had already been partially reported in another study using the same methodology [74]. From the clear results found, an important issue emerges: can these be two different sources of probability of occurrence of a sports injury?

It should not be forgotten, on the other hand, that this characteristic of seeking out and undertaking risky behavior may have some bearing on the issue—which is critical on many levels today—of sports betting. This would seem to be confirmed by the fact that the highest values for the factors in this variable correspond to financial, social, and sport/leisure factors. This separation of the two nodes opens, in our opinion, an important collateral avenue for furthering our understanding of the psychological components of sport injuries.

Furthermore, the findings of this study can—as indicated, in part, above—fit very well into the proposed integrative model (the GMSI) [36] as they can be considered as antecedents of injury and, therefore, enter into the realm of possible injury prevention, rather than rehabilitation or recovery and the concept of ‘returning to the sport’.

When we analyze the results of the instantiations on the bottom (or consequent) variables with hypothetically unachievable maximum and/or minimum values, the results found in the BN are clearly reaffirmed.

The results obtained can contribute to both theoretical and applied aspects of the scientific problem. On the one hand, from a theoretical perspective, the inclusion of “anxiety” and “depression” at the level of “stress” in the model of Andersen and Williams may allow a conceptually different approach, being able to speak of a “negative triad”, and the need for empirical studies that can support this aspect in line with the proposals of Olmedilla and García-Mas in their GMSI [36]. On the other hand, the results seem to show that the presence of mental health indicators in sports practice is a fact of great importance, and not only from the perspective of injury [75]. In this sense, it seems necessary to implement psychology actions and programs for the promotion of mental health, and for the prevention of the basic indicators of the “negative triad”, in line with Brenner et al. [76] and Henriksen et al. [77].

## 5. Conclusions

First of all, the existence of the “triad” node is consolidated, as the variables contained in it are affected jointly, with an almost “homogeneous” function (possibly, both in the occurrence and in the rehabilitation process of the injury) on modifying the stress values upwards. In addition, the separation of increase/decrease of the occurrence values that may hypothetically reach the probability of the risk tendency is maintained.

Likewise, we also see that a variable that is much discussed in many aspects and theoretical frameworks (see, for example, the corresponding scale of the updated sport engagement questionnaire, the SCQ-2 [78]), social support, is highly compromised both in the BN and, particularly, in the instantiations, as its probability of occurrence does not increase when the “negative” variables do. It is true that it has previously been pointed out within different areas that the line between support and social pressure is not entirely clear-cut and immovable [79,80], and this seems to be another example of it. As such, what seems to happen when the probability of occurrence of the SS decreases, is that it leads to an increase in the other factors (D, A and E) of the same variable, and a decrease in the probability values of risk-tendency factors. Perhaps the fact of not having a ‘safety net’ makes the players implement strategies and resources with greater implication (“...there is no family or anyone to protect me, I have to get on with it”, “...either I do it or...I don’t have anyone”), which may help to have greater focus, decide more quickly and stabilize behavior, along the lines of what is foreseen in the stress model.

And this immediately opens up another very relevant question, especially regarding youth sport: which psychosocial factor can be more, and which can be less, effective in preventing or aiding recovery from a sports injury? In this case, we fear that we will have to resort to methodologies of a more qualitative nature to complement the quantitative ones in order to obtain answers adapted to the specific characteristics of the athletes. As in our study, it has been shown that the family “umbrella” does not seem to have relevant probabilistic weight.

Finally, based on the results obtained and on this critical discussion, it can be stated that stress is not only found within the framework of psychological variables traditionally considered as “negative”, and that these have different degrees of severity and complexity for the players. It can also be affirmed that intervention plans should separate the intervention on these psychological components from those related to the learning of avoidance and risk behavior.

As a last conclusion, we can state that stress is not the only factor in the classic and repeatedly proven model linking it to sports injuries.

### 5.1. Limitations and Future Developments

Obviously, there are some limitations in this study. Firstly, there is a transversal analysis, due to the rapid and dynamic composition of sports teams, which see changes regarding some of their players almost every season. This classic problem when working with sport competition teams is really difficult to address, hindering somewhat the longitudinal approach. Secondly, there is no work on the best way to better integrate the classic statistical methods with the Bayesian analysis (we are currently conducting them in a sequential way, gaining information on one over the other).

Thirdly, regarding the ad hoc questionnaire, its design does not provide enough specific information (is it a new injury, a relapse of an old one or another injury different from the one previously sustained?); the place of the injury (practice, competition or off the pitch); time the injury took place (opening the door to the study of the chronobiological factors in this field), and, in a more psychological fashion, its design must include the causal attribution made by the injured athlete about the cause of the injury.

Finally, the authors didn’t have the opportunity to study the physical and body indications associated with the stress present in some players. This kind of evaluation may be a valuable complement to the paper and pencil data collection, when studying the anxiety and stress associated with the injury.

Regarding future developments, there are three lines of work in the medium term. Firstly, and given the results obtained, it would be interesting to compare the graphs and BNs of the two semi-samples of this study (injured vs. non-injured players) in order to observe the differences, if any, between the variables, their nodes and their probabilities of occurrence.

Secondly, once this extension of the stress model is “secured”, it may also be relevant—in this case, following the GMSI conceptual model—whether depression is pre-injury (antecedent) or post-injury (consequent) and, in this case, we will have to work with a longitudinal follow-up or survival methodology and sample.

Thirdly, it would be good to improve our knowledge of the injuries and specific features, working with bigger samples including indoor football and female teams, regarding some particularities discovered in the kind of injuries sustained.

Further studies should also look into the differences between different sports and the injuries that occur, following, on the one hand, the classical differentiation between individual and team sports, as well as direct and indirect opposition sports. Furthermore, the analysis between more socially developed sports (such as biking and trekking), which currently take place in a very high percentage outside the competitive area and are done by amateurs, should be developed (this has not been systematically conducted so far). This type of analysis could—along the lines of the BN analyses—indicate different probabilities of sustaining an injury associated with doing different sports.

Finally, the role of competitive anxiety in relation to many sporting situations is very controversial indeed [46]. In this case, a line of study and development regarding anxiety (three-dimensionally considered) in relation to “returning to the sport” may be of great value from a theoretical and applied intervention perspective. This study should perhaps be carried out using a qualitative rather than quantitative methodology.

### 5.2. Practical Implications

Despite the limitations indicated above, this study has high practical implications for the professional practice of both sports coaches and other support professionals (sports doctors, sports psychologists, physical trainers, physiotherapists, etc.), given that traditional descriptive analyses indicate a trend in the results according to the sports age category (but not according to the sex category), thus we know that this is a variable that should be addressed in the pre- and post-injury intervention. Likewise, the analyses developed through BNs indicate the importance of generating adequate coping strategies in athletes (especially considering the specificity of possible social support at the sporting and extra-sporting level), with a view to preventing and coping with two possible consequences of a failure in the adaptation process of athletes: anxiety and depression.

## Figures and Tables

**Figure 1 healthcare-10-00236-f001:**
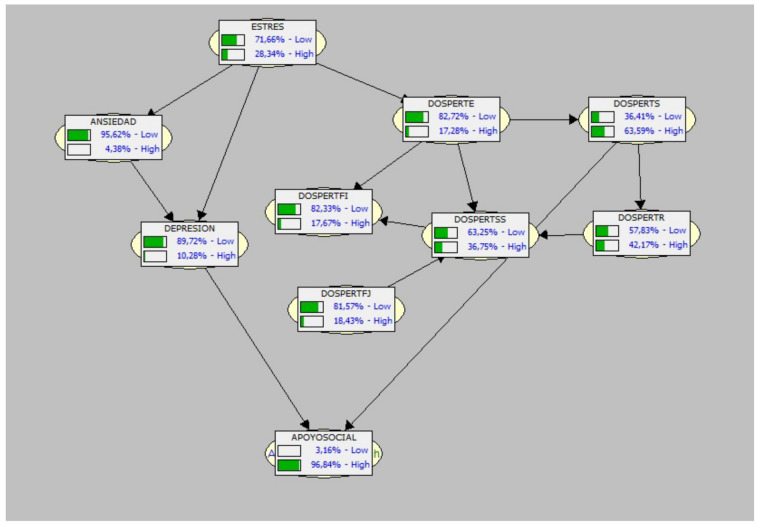
Graph and Bayesian Network discovered with values (high and low probability) of the study variables. Legend: Dospert: E (Ethical); FI (Financial/Business); FJ (Financial/Betting); SS (Health/Safety); S (Social); R (Recreational/Sport).

**Table 1 healthcare-10-00236-t001:** Table of contingency of the sample considering the variables of sex and type of sport.

			Sport Played		Total		
			Football	Indoor Football		X Age	SD
Sex	Male	Number	300	23	323	23.66	4.36
		% of the total	65.9%	5.1%	71%		
	Female	Number	99	33	132	19.61	4.00
		% of the total	21.8%	7.3%	29%		
Total		Number	399	56	455	21.66	4.18
		% of the total	87.7%	12.3%	100.0%		

**Table 2 healthcare-10-00236-t002:** Table of contingency of the sample considering the variables of number of injuries, sex and type of sport.

Sex	Sport		Number of Injuries						Total
			0	1	2	3	4	5	
Male	Football	Number	155	120	20	4	0	1	300
		% of the total	48.0%	37.2%	6.2%	1.2%	0.0%	0.3%	92.9%
	Indoor football	Number	16	6	0	0	1	0	23
		% of the total	5.0%	1.9%	0.0%	0.0%	0.3%	0.0%	7.1%
	Total	Number	171	126	20	4	1	1	323
		% of the total	52.9%	39.0%	6.2%	1.2%	0.3%	0.3%	100.0%
Female	Football	Number	46	42	10	1			99
		% of the total	34.8%	31.8%	7.6%	0.8%			75.0%
	Indoor football	Number	18	13	2	0			33
		% of the total	13.6%	9.8%	1.5%	0.0%			25.0%
	Total	Number	64	55	12	1			132
		% of the total	48.5%	41.7%	9.1%	0.8%			100.0%
Total	Football	Number	201	162	30	5	0	1	399
		% of the total	44.2%	35.6%	6.6%	1.1%	0.0%	0.2%	87.7%
	Indoor football	Number	34	19	2	0	1	0	56
		% of the total	7.5%	4.2%	0.4%	0.0%	0.2%	0.0%	12.3%
	Total	Number	235	181	32	5	1	1	455
		% of the total	51.6%	39.8%	7.0%	1.1%	0.2%	0.2%	100.0%

**Table 3 healthcare-10-00236-t003:** Descriptive and mean difference analysis applying Student’s *t*-statistic for two independent samples considering the variables: number of injuries, sport category per sex, type of sport and age of the players.

		*N*	*M*	*SD*	*F*	Sig.	*T*	Sig.	Cohen’s d
Sex category	Male	323	0.58	0.737	0.072	0.789	−0.567	0.571	−0.056
	Female	132	0.62	0.683					
Sport	Football	399	0.61	0.718	0.218	0.641	1210	0.227	0.121
	Indoor football	56	0.48	0.738					
Age	>18 years old	75	0.45	0.664	0.819	0.366	−1821	0.069	0.244
	≤18 years old	378	0.62	0.730					

**Table 4 healthcare-10-00236-t004:** Validation of the obtained BN through AUC indicators.

	AUC	Accuracy	Sensitivity	Specificity
STRESS	0.6	0.75	0.95	0.23
ANXIETY	0.5	0.96	1	0
DEPRESSION	0.6	0.92	0.99	0.18
DOSPERTE	0.63	0.83	0.94	0.32
DOSPERTFI	0.62	0.85	0.99	0.26
DOSPERTFJ	0.58	0.8	0.93	0.22
DOSPERTS	0.7	0.68	0.77	0.63
DOSPERTSS	0.64	0.72	0.92	0.36
DOSPERTR	0.66	0.67	0.72	0.6
SOCIAL SUPPORT	0.5	0.97	0	1

**Table 5 healthcare-10-00236-t005:** Instantiation 1. Social Support, from 96.64% to 0% HIGH Likelihood. Instantiation 2. Depression, from 10.24% to 100% HIGH Likelihood. Instantiation 3. DESPORT SS, from 36.75% to 100% HIGH Likelihood. Instantiation 4. DESPORT SS, from 36.75% to 100% HIGH, Depression, from 10.24% to 100% HIGH, and Social Support, from 96.63% to 0% HIGH, likelihoods simultaneously.

	Social Sup.	Depression	Anxiety	Stress	DE	DS	DFI	DFJ	DR	DSS
Original BN										
HIGH	96.63	10.24	4.38	28.34	17.28	63.59	17.67	18.43	42.17	36.75
LOW	3.16	89.72	95.62	71.64	82.72	36.41	82.33	81.57	57.43	63.25
1. After Social Support = 0% High										
HIGH	0	44.41	14.09	39.38	16.21	50.15	17.22	18.43	36.46	35.09
LOW	100	55.59	85.91	60.62	83.79	49.85	82.78	81.57	63.54	64.91
2. After Depression = 100% HIGH, 0% LOW										
HIGH	86.91	100	30.39	55.97	20.02	64.27	18.44	18.43	35.09	37.92
LOW	13.09	0	69.61	44.03	79.98	35.73	81.56	81.57	64.91	62.08
3. After Dosport SS = 100% High, 0% LOW										
HIGH	96.98	10.66	4.61	30.73	34.23	72.8	30.2	24.58	57.94	100
LOW	3.02	89.34	95.39	69.27	65.77	27.2	69.8	75.42	42.06	0
4. After Dosport SS = 100% High, 0% LOW; Depression = 100% HIGH, 0% LOW, and Social Support = 0% HIGH, 100% LOW										
HIGH	0	100	30.07	58.17	41.34	87.3	32.152	3.46	64.18	100
LOW	100	0	69.93	41.87	58.66	12.7	67.857	6.54	32.82	0

## Data Availability

Data supporting the reported results are found in Department of Personality, Evaluation, and Psychological Treatment, Faculty of Psychology, Murcia. Responsible is Aurelio Olmedilla Zafra.

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
