# Peer review of "It Is Not Just Stress: A Bayesian Approach to the Shape of the Negative Psychological Features Associated with Sport Injuries"

_healthcare, 2022, doi:10.3390/healthcare10020236_

Round 1
Reviewer 1 Report
Some global comments:
If the study objective is to extend the stress and injury model of Andersen and Williams, to other “negative” psychological variables, such as anxiety associated to competition and depression, encompassed in the conceptual model of Olmedilla and García-Mas. The research goal shouldn’t be the prediction of injuries using this new variables? Again, where is the analysis and comparison between the Anderson Model vs Olmedilla Model?
Therefore, I think this should be better explained and the objectives more aligned with the analysis developed, where the t-students and contingency tables are supporting what? Considering the research objectives.
After is stated that “Therefore, based on the studies carried out and the results shown in Tables 2 and 3, 270 we can see that no statistically significant differences have been found (and that the trends 271 have a small effect size) among the various variables in the study.” Where this is was considered in the research aims and why is this important? Therefore, could be used for example the retrieved information and for example a logistic regression to predict the injuries (yes/no), considering the research objectives.
The Model presented in figure1 should be developed also for the injury model with less information (Anderson) and confronted with Olmedilla model.
The objectives should be better aligned with the analysis developed and subsequent discussion, it seems that the results and discussion are not related to the research objectives.
Author Response
Dear reviewer, we appreciate your comments to improve the quality of the article.

Reviewer 2 Report
Thank you for the opportunity to review this interesting paper. In general, this manuscript addresses a critical topic and is well-organized. However, there are still some issues that need to be addressed for re-submission.
First, the authors need to put more effort to link this study to the scope of this journal. That is, how athletes' injuries can be relevant to "healthcare"? Doing so can make this study more suitable for this journal. Moreover, more significance and importance of sports injuries for athletes need to be addressed. Sports injuries affect athletes in many different aspects. Stronger justification for sports injuries is necessary for this study.
Second, the demographic information of participants is not clear. Negative psychological outcomes may differ at different levels of sport participation. For example, it has been found that athletes at a higher level may have more psychological difficulties during transitions (including injuries) (e.g., Samuel &Tenenbaum, 2011).
Third, the findings are interesting and valuable. However, the authors need to detail how the new findings can contribute to the theoretical and practical implications. I believe the findings of this study can significantly contribute to the coaching practice and athletes' well-being.
In addition, usually, the conclusion should be the last section and include only the answers to the research questions and vital points of this study. The limitations and implications should be within the section of the Discussion. Please re-organize the final sections of the manuscript.
Samuel, R. D., & Tenenbaum, G. (2011). The role of change in athletes’ careers: A scheme of change for sport psychology practice. The Sport Psychologist, 25(2), 233-252.
Author Response

(The authors gave the same response as above.)

Reviewer 3 Report
Congratulations to the authors.
It is a good study, rigorous in methodology, using an innovative analysis that is increasingly used in the field of psychology research in sports sciences. Its results are quite interesting, giving a new approach to the evolution of research in this field.
Regarding the strengths found in the article, it is worth highlighting the importance of carrying out this type of control experimentation in the different studies and investigations that the authors carry out in sport psychology, where two different and different models are related (Andersen's and Williams Versus that of Olmedilla and García Mas) through negative psychological variables (such as anxiety and depression) in the field of knowledge of psychological aspects of sports injuries.
On the other hand, also the inclusion and use of classical statistical methods in the investigation of psychological processes together with other more innovative ones such as Bayesian networks. As I told you in the previous post, I am going to start to carry out my studies on this type of models to investigate what results I get and to be able to make a good discussion.
Regarding the weaknesses, perhaps they should have carried out this type of study with other samples of different athletes in relation to sex and age types, implementing longitudinal designs in the process.
On the other hand, the inclusion of new negative variables in confrontation with a positive one could give a different approach and another perspective to the study, with interesting results.
Author Response

(The authors gave the same response as above.)

Reviewer 4 Report
As a wish to the authors. Sports injury is, of course, associated with the psychological state of the athlete, including stress. However, the occurrence of trauma is not limited only to the psychological state, there are other reasons.
My criticisms are related to limiting work results. I think that authors must consider the different factors which are related to sports injury. If authors include the other factors and will show relation with stress this manuscript will be accepted.
Author Response

(The authors gave the same response as above.)

Reviewer 5 Report
- Abstract: “An ad hoc questionnaire was used for the corresponding sociodemographic data and data relating to injuries. The DASS-21 Questionnaire was used to assess depression, anxiety and stress. The DOSPERT-S Questionnaire was used to assess the risk tendency. The 26 Multidimensional Scale of Perceived Social Support was used to assess social support.” I think it is not necessary to name the survey instruments here, just tell readers the variables you measured/studied
-Line 42: typo?
-Line 65: I believe there are also other models going beyond the GPMSIs, it would be great to give readers a fuller picture around here before you narrow down your review or discussion to this specific model. For example: Li C, Ivarsson A, Lam LT, Sun J. Basic psychological needs satisfaction and frustration, stress, and sports injury among university athletes: a four-wave prospective survey. Frontiers in Psychology. 2019 Mar 26;10:665.
-Line 94: Research hypotheses are lacking?
-Line 110: “Risk-taking in specific context” seems new and sudden as I did not aware you have discussed any related in your Introduction?
-Line 114-115: Just be straightforward, do you mean a convenient sample? It is unclear how n>400 is reached to make sure the sampling error is below 5%?
-Line 119: Spell out “DT”. Probably include age information in Table 1 and decrease your age description here
-Line 122-125: Avoid reporting the same information using both text and table?
-Line 131-133: Unnecessary information, consider removing this paragraph?
-Line 138-145: Need a reference here?
-Line 146-179: Try to simplify these instrument descriptions a bit particularly the psychometric properties information
-Line 219-223: Consider removing this paragraph
-Table 2: Again in your main text try not to repeat too much information that has been presented in Table 2
-Table 3: Only one d value in your table, what about the “sex” and “sport” categories?
-Line 280-282: Cut this paragraph out?
-Table 5-8 can be combined as one table
-Line 380: What are these variables? Name them?
-Line 435: I think this section should be placed at the last rather than here? The whole section is too long; a conclusion section is supposed to convey the key information/finding after your discussion
Author Response

(The authors gave the same response as above.)

Round 2
Reviewer 1 Report
I would like to thank the authors for addressing the questions and making improvements.
Reviewer 5 Report
Thank you for addressing my comments. I have no further comments.